# Antibiotic Prescribing by Informal Healthcare Providers for Common Illnesses: A Repeated Cross-Sectional Study in Rural India

**DOI:** 10.3390/antibiotics8030139

**Published:** 2019-09-05

**Authors:** Shweta Khare, Manju Purohit, Megha Sharma, Ashok J. Tamhankar, Cecilia Stalsby Lundborg, Vishal Diwan, Ashish Pathak

**Affiliations:** 1Department of Public Health Sciences, Global Health—Health Systems and Policy (HSP): Medicines Focusing Antibiotics, Karolinska Institutet, Tomtebodavagen 18A, SE-171 77 Stockholm, Sweden; 2Department of Public Health and Environment, RD Gardi Medical College, Ujjain, Madhya Pradesh 456006, India; 3Department of Pathology, RD Gardi Medical College, Ujjain, Madhya Pradesh 456006, India; 4Department of Pharmacology, RD Gardi Medical College, Ujjain, Madhya Pradesh 456006, India; 5Indian Initiative for Management of Antibiotic Resistance, Department of Environmental Medicine, RD Gardi Medical College, Ujjain, Madhya Pradesh 456006, India; 6International Centre for Health Research, Ujjain Charitable Trust Hospital and Research Centre, Ujjain, Madhya Pradesh 456001, India; 7Department of Pediatrics, RD Gardi Medical College, Ujjain, Madhya Pradesh 456006, India; 8Department of Women and Children’s Health, International Maternal and Child Health Unit, Uppsala University, SE-751 85 Uppsala, Sweden

**Keywords:** Healthcare providers, prescription, antibiotics, infectious diseases, rural population, India

## Abstract

Informal healthcare providers (IHCPs) are predominant healthcare providers in rural India, who prescribe without formal training. Antibiotic prescription by IHCPs could provide crucial information for controlling antibiotic resistance. The aim of this study is to determine the practices and seasonal changes in antibiotic prescribing for common illnesses by IHCPs. A repeated cross-sectional study was conducted over 18 months, covering different seasons in the rural demographic surveillance site, at Ujjain, India. Prescriptions given to outpatients by 12 IHCPs were collected. In total, 15,322 prescriptions for 323 different complaint combinations were analyzed, of which 11,336 (74%) included antibiotics. The results showed that 14,620 (95%) of antibiotics prescribed were broad spectrum and the most commonly prescribed were fluoroquinolones (4771,31%), followed by penicillin with an extended spectrum (4119,27%) and third-generation cephalosporin (3069,20%). Antibiotics were prescribed more frequently in oral and dental problems (1126,88%), fever (3569,87%), and upper respiratory tract infections (3273, 81%); more during the monsoon season (2350,76%); and more frequently to children (3340,81%) than to adults (7996,71%). The study concludes that antibiotics were the more commonly prescribed drugs compared to other medications for common illnesses, most of which are broad-spectrum antibiotics, a situation that warrants further investigations followed by immediate and coordinated efforts to reduce unnecessary antibiotic prescriptions by IHCPs.

## 1. Introduction

Antibiotic resistance (ABR) has become a major threat to global health and is regarded as a complex multidimensional crisis [1]. ABR adversely affects healthcare, and thus requires immediate, concerted, global action [1]. Antibiotics have been used indiscriminately around the world, but the problem of inappropriate antibiotic use is rampant in low- and middle-income countries (LMICs) [1,2]. Although the development of ABR is a natural evolutionary process, the indiscriminate use of antibiotics in humans has accelerated it; antibiotic use in domesticated animals has resulted in the entry of antibiotics into the food chain and environment [3]. Multiple factors are responsible for human ABR in LMICs, including the high burden of infectious diseases, poor hygiene and infection control measures, the easy availability of antibiotics, the indiscriminate use of antibiotics, the poor quality of diagnostics, the lack of availability of treatment guidelines, non-adherence to treatment guidelines, and the diverse and fragmented healthcare systems [4,5,6,7].

India has a large three-tier public healthcare system that ensures easy access to primary care, irrespective of the socioeconomic condition of an individual; however, this system has been ineffective in providing primary care in rural areas [8]. One of the reasons for failure of the public healthcare system is the shortage of medical staff (0.39 qualified doctors per 1000 people in rural areas vs. 1.33 qualified doctors per 1000 people in urban areas) and the prolonged absence of appointed medical staff [9,10]. Failure of the public healthcare system has resulted in the evolution of private healthcare systems [8], which cater to 78% and 60% of outpatients and inpatients, respectively, in India [11]. In many Asian countries where healthcare systems are weak, healthcare tends to shift into the hands of informal healthcare providers (IHCPs) [12]. IHCPs are defined as healthcare providers who have not received a formal degree in medicine from any institution and are not registered as healthcare practitioners with any governing body. While some may have received some informal training, they are not certified by any formal institute [12]. In India, IHCPs include unqualified doctors, spiritual healers, unqualified drug vendors, and traditional birth attendants [9,13,14,15], and this informal cadre constitutes more than half of all active healthcare providers in rural India [9]. Most IHCPs prescribe and dispense allopathic medicines; only a few practice pure traditional cures [16]. Most medications, including antibiotics, are readily available for IHCPs to prescribe and administer [14], due to the poor enforcement of regulations on their distribution. Seasonality in the prescription and consumption of antibiotics has been assumed to influence antibiotic resistance in bacteria [17]. Seasons influence health-seeking at the community level, resulting in spikes in the indiscriminate use of unnecessary antibiotics that generate seasonal patterns of resistance [18]. Coordinated actions at the community level are needed in order to control unnecessary antibiotic use [18].

Studies quantifying and explaining the antibiotic prescribing practices of IHCPs in India are not currently available. In the present study, the antibiotic prescription patterns of IHCPs for common illnesses in rural Ujjain, India, were explored by conducting repeated follow-ups over 18 months and by identifying seasonal changes in prescription patterns.

## 2. Results

A total of 15,488 prescriptions were collected from IHCPs in 18 months, but 166 (1%) prescriptions had incomplete information on age, presenting complaints, and were not included in the analysis. Therefore, 15,322 prescriptions of patients with 323 different complaints (either alone or in a combination of two or three complaints) were included in the study. All the IHCPs who participated in the study were men. Nine IHCPs had clinics to examine and treat their patients; the remaining three visited the homes of their patients on call and did not have fixed clinics. The median age of the patients was 30 years (Inter Quartile Range 15–45, range 14 days–100 years), with 57% male and 43% female patients. A total of 4099 (27%) prescriptions were for children. Out of 15,322 prescriptions, 14,074 (92%) prescriptions were with presenting complaints having a frequency higher than 100. The more common presenting complaints identified in the study were fever (4118, 27%), upper respiratory tract infection (URTI) (4047, 26%), gastro-intestinal disorders (2107, 14%), oral and dental problems (1273, 8%), skin infections (1068, 7%), unspecified pain (785, 5%), injury (322, 2%), and asthma (354, 2%). Analyses of these presenting complaints are presented in detail in Table 1, Table 2, and Table 3, and Figure 1.

### 2.1. Antibiotic Prescribing for Presenting Complaints

Antibiotics were prescribed in 74% (11,336/15,322) of prescriptions. Overall, 11,336 prescriptions contained 15,472 antibiotics prescribed either singly or in combination of upto five antibiotics (mean ± standard deviation (SD) 1.4 ± 0.5 antibiotics per prescription). Antibiotics were prescribed more frequently to children under 5 years of age (85%; 1899/2247) than to adults (71%; 7996/11223) and more frequently to males (75%; 6507/8679) than to females (73%; 4829/6643) (Table 1). Parenteral formulations were prescribed less often than oral formulations (Table 2) but were more likely to be prescribed to the age group 6–17 years (17%; 855/5007) vs. age group ≤5 years (15%; 742/5007).

The majority (95%; 14620/15472) of antibiotics prescribed were broad spectrum and the most commonly prescribed were fluoroquinolones (J01MA) (*n*=4771; 31%), followed by penicillin with an extended spectrum (J01CA) (*n*=4119; 27%) and third-generation cephalosporin (J01DD) (*n*=3069; 20%). Specially, ofloxacin (*n*=2723; 17.6%), amoxicillin (*n*=2583; 16.7%), cefotaxime (*n*=1851; 12%), ampicillin cloxacillin combination (*n*=1530; 9.9%), ciprofloxacin (*n*=1502; 9.7%), and gentamicin (*n*=1475; 9.5%) represented 75.4% of all prescribed antibiotics (Table 3 and Figure 2). Antibiotics were prescribed more frequently in oral and dental problems (88%; 1126/1273) and fever (87%; 3569/4118) compared to skin infections (79%; 844/1068) (Table 1).

Seasonal variations were observed in both the frequency of the complaints reported by the patients and the corresponding antibiotic prescribing (Appendix A and Figure 1). The peak of antibiotic prescribing was observed during the monsoon seasons: both monsoon seasons (76%) (Table 1 and Figure 1). Significant seasonal variations were observed in antibiotic prescription rates for URTIs and gastro-intestinal disorders, and peaks were noted during winter and the first pre-monsoon season, respectively (Figure 1).

#### 2.1.1. Fever

A total of 4118 (27%) prescriptions had fever as the presenting complaint, out of which 87% (3569/4118) were prescribed antibiotics (Table 1). Overall, fluoroquinolones were the most common class prescribed, followed by third-generation cephalosporins (Table 3). The most common antibiotics prescribed to children were amoxicillin (J01CA04; 316 Defined Daily Doses (DDD)/ 1000 prescriptions) and cefotaxime (J01DD01; 66 DDD/ 1000 prescriptions), whereas ofloxacin (J01MA01; 1079 DDD/ 1000 prescriptions) and gentamicin injection (J01GB03; 655 DDD/ 1000 prescriptions) were prescribed to adults (Table 3 and Figure 2).

#### 2.1.2. Upper Respiratory Tract Infection (URTI)

A total of 4047 (26%) prescriptions were with URTIs, and 81% (Table 1) of them were prescribed antibiotics. The most commonly prescribed antibiotics to children were amoxicillin (J01CA04; 709 DDD/ 1000 prescriptions) and cefotaxime (J01DD01; 61 DDD/ 1000 prescriptions). In adults, the most commonly prescribed antibiotics were ofloxacin (J01MA01; 1093 DDD/ 1000 prescriptions) and cefotaxime injection (J01DD01; 135 DDD/ 1000 prescriptions) (Table 3 and Figure 2).

#### 2.1.3. Gastro-Intestinal Disorders

Gastro-intestinal disorders were the presenting complaint in a total of 2107 (14%) prescriptions (Table 1); of this, 15% were in the age group ≤5 years. Higher proportions of antibiotics were prescribed to the age group ≤5 years (70%; 222/318) compared to the age group ≥18 years (59%; 934/1589) (Appendix A). Overall, aminoglycosides were the most commonly prescribed group through the parenteral route. In children, the most common antibiotics prescribed were a fixed-dose combination (FDC) of norfloxacin and metronidazole (J01RA; 482 DDD/ 1000 prescriptions) and amikacin injection (J01GB06; 113DDD/ 1000 prescriptions). In adults, amikacin injection (J01GB06; 419 DDD/ 1000 prescriptions) was the most prescribed antibiotic, followed by FDC of norfloxacin and tinidazole (J01MA01; 1555 DDD/ 1000 prescriptions) (Table 3 and Figure 2).

#### 2.1.4. Oral and Dental Problems

Very few (5/1273, Appendix A) prescriptions to children were reported to have oral or dental problems. About 90% of adult patients having oral or dental problems received antibiotics. For adults, the most common antibiotics prescribed were FDC of ampicillin with cloxacillin (J01CR50; 3661 DDD/ 1000 prescriptions), together with ofloxacin (J01MA0; 1093 DDD/ 1000 prescriptions) (Table 3 and Figure 2).

#### 2.1.5. Skin Infections

Skin infection was the diagnosis included in 1068 (7%) prescriptions (Table 1). Of the children presenting with skin infections, 86% were prescribed antibiotics, compared to 77% of adults (Appendix A). Ciprofloxacin (J01MA02; 945 DDD/ 1000 prescriptions) and gentamicin injections (J01GB03; 709 DDD/ 1000 prescriptions) were the commonly prescribed antibiotics in both adults and children (Table 3 and Figure 2).

#### 2.1.6. Unspecified Pain

Among the prescriptions for pain as the presenting complaint (site of pain was unspecified), 30% of the adults were prescribed antibiotics (Appendix A). Ciprofloxacin (J01MA02; 833 DDD/ 1000 prescriptions) and ofloxacin (J01MA01; 1061 DDD/ 1000 prescriptions) were the commonly prescribed antibiotics (Table 3 and Figure 2).

### 2.2. Cluster Analysis

The heat map for the clustering of antibiotic prescribing according to diagnosis is shown in Appendix A. The F-ratio for the variance of antibiotic prescribing by prescriber was 2.31 (*p* = 0.0154) and variance by diagnosis was 8.089 (*p* < 0.0001). Therefore, there is significant variation in antibiotic prescribing by both prescriber and diagnosis, but the variation is greater by diagnosis. The variation is also clearly visible in the heat map (Appendix A).

## 3. Discussion

To the best of our knowledge, this is the first study to describe and quantify the antibiotic prescription practices of IHCPs in rural India, contrasting with other studies in India conducted in a hospital setting with formal healthcare providers [19,20,21,22,23,24,25]. The results of our study reveal that the IHCPs prescribed antibiotics at a high proportion (74%) for common illnesses compared to other non-antibiotic medications. Although the proportion was high across all three age groups, the proportion of patients receiving antibiotic prescriptions was higher in the age group ≤5 years (0.85; 95% confidence interval (CI) 0.83–0.86) than in the age group ≥18 years (0.71; 95% CI 0.70–0.72). Notably, the total number of prescriptions in the age group ≤5 years was lower than that in the age group ≥18 years (27% vs. 83%, respectively). The overall antibiotic prescription rate in the present study was higher than that reported in other studies evaluating outpatient antibiotic prescription rates in hospital settings [19,20,23]. Antibiotic prescription rates were relatively high in the age group ≤5 years in the present study compared to other studies in India [19,20,23]. A study conducted in Uttar Pradesh, India, reported an antibiotic prescription rate of 82% in outpatients in urban private and rural settings for formal healthcare providers [19,20,23].

The study revealed seasonal variation in overall antibiotic prescribing rates, with a peak rate occurring in the monsoon season (Table 1 and Figure 1). Similar seasonal variations were reported in a previous study from the same setting, which showed a peak in antibiotic prescription rates during the monsoon season [23]. A study conducted in the city of New Delhi, India, reported similar results, that a higher proportion of patients were prescribed antibiotics for URTI and diarrhea during winter and humid summer months [22]. The prescription patterns of the IHCPs, particularly for loose motions, indicated that they probably duplicated treatments prescribed by qualified medical practitioners from the same area [24]. Similar findings were reported in a qualitative study on IHCP treatment practices [25].

Patients presenting with URTIs and gastro-intestinal disorders, which are more often caused by viral infections, were prescribed antibiotics (Table 1) [6]. According to the recommendations of the Indian National Treatment Guidelines for Antimicrobial Use in Infectious Diseases, amoxicillin or macrolides is the first-line empirical treatment for lower respiratory tract infections, and first-generation cephalosporins, if a patient exhibits penicillin allergy; antibiotics are not indicated for URTIs [6]. In our study, we found that most patients with a diagnosis of URTI received prescriptions for third-generation cephalosporins and fluoroquinolones (Table 3), which is inappropriate. Similarly, for fever, the most commonly prescribed antibiotics were fluoroquinolones, macrolides, and third-generation cephalosporins (Table 3); these prescriptions are not in accordance with treatment guidelines [6]. At the individual patient level, the consequence of an unsuitable antibiotic prescription has been reported to be associated with a two-fold higher risk of resistance to the prescribed antibiotic and other adverse effects [26]. Excessive and unindicated use of broad-spectrum antibiotics is also a cause of concern because it propagates ABR, not only among pathogenic bacteria [27], but also among commensal bacteria [28]. In the present study, the overall prescription rate of fixed-dose combinations was 16% (Table 2). Injectable antibiotics were prescribed in 5007 (32%) prescriptions. These injectable and oral drugs are dispensed by the IHCPs, mainly with the motive of making profit by charging a mark-up over the distributor prices [29]. This high prescription rate and inappropriate use of antibiotics by the IHCPs may be due to a lack of knowledge and training, an inadequate understanding of the use of antibiotics against viruses, a lack of knowledge of differential diagnoses of diseases and resulting diagnostic uncertainty, lack of diagnostic tests prescribed, perceived patient expectation for antibiotics, prevailing culture, fear of losing a patient to other IHCPs, and the belief that antibiotics are magic bullets [3]. However, it has been shown in other studies that training improves the case management of the illnesses but might not decrease antibiotic prescribing [29].

### 3.1. Methodological Considerations

One of the strengths of this study is that we could collect antibiotic prescription data from the IHCPs. Our data shows that IHCPs should be considered important component in future effective antibiotic stewardship interventions. However, we could only survey 12 IHCPs. Given the large number of active IHCPs in the study area, there is risk of selection bias. Additionally, as the prescriptions were written by IHCPs themselves, there is a risk of observation bias. Both selection and observation bias would deflate the antibiotic prescribing rate, this is important in view of the high antibiotic prescribing rate observed in the study. Due to the small number of participants, we could not study differences in prescribing patterns by IHCP characteristics, such as age and/or gender. These provider characteristics are of interest and require further investigation. Coding of the prescriptions was done under the supervision of the medical doctor and took a significant amount of time as each prescription included variable units for prescribed antibiotics. For calculating DDDs, not only the formulations for tablets, syrups, and injections were taken into account, but a significant amount of time was also spent on standardizing the unit of measurement, for example, gram, milligram, milliliter, one vial, or half vial of injection, etc. were all changed to grams. Furthermore, the DDD methodology does not provide pediatric doses. However, since occasions of prescribing are reported throughout the study, this problem of pediatric DDD is minimized.

### 3.2. Strengths and Limitations

This study presents the following strengths and limitations:This study is the first to describe and quantify the antibiotic prescription practices of IHCPs in rural India, although with some difficulty, due to the illegal status of their practice;We could only collect prescription data from 12 IHCPs, as many of the non-participants were afraid to reveal information regarding their treatment practices;Repeated data collection, including monthly recording of a considerable number of antibiotic prescribing occasions from the same group of IHCPs, made the data reliable and this study design also enabled us to minimize the percentage of missing data (1% missing data overall);We assessed seasonal changes in antibiotic prescription rates, which has rarely been reported in studies from India;The study did not evaluate the severity of the disease.

### 3.3. Recommendations

Combating ABR is now a high priority [30]; this study underscores the immediate need for coordinated actions at the policy-making level. Interventions are needed to prevent irrational antibiotic prescriptions by IHCPs. IHCPs could be trained and certified as recognized healthcare workers, which could facilitate their integration into the legal healthcare system and help policy makers increase the accessibility of healthcare. One of the immediate and coordinated actions could be ‘task shifting,’ as proposed by WHO, which involves the rational redistribution of specific tasks from highly qualified health workers to health workers with relatively little or no training, such as IHCPs [31]. Recently, the Government of India decided to upgrade the skills of IHCPs by providing short-term training or courses under the flagship program ‘skill development’ [32]. We are hopeful that this may help reduce the inappropriate and excessive antibiotic prescribing, with a consequent reduction in the incidence of antibiotic resistance.

## 4. Materials and Methods 

### 4.1. Study Design and Study Setting

This repeated cross-sectional study was conducted in the Ujjain district of the state of Madhya Pradesh (MP), India. MP has a low human development index of 0.624 [33]. It is the fifth largest state in India in terms of population, and approximately 75% of its population resides in the rural area [34]. Specifically, the study was conducted in the rural demographic surveillance site (DSS), Palwa, of RD Gardi Medical College, Ujjain, which caters to 60 villages in three development blocks (Figure 3) [35]. The populations of each of the 60 DSS villages’ ranges from 800 to 2500, and approximately 50% of the villages have a population of less than 1000 inhabitants. Most (80%) of the healthcare providers in the DSS, both formal and informal, are concentrated in one village, known as the central village (Figure 3).

### 4.2. Selection of Study Participants

The area within a 5km radius of the central village formed the sampling frame for the selection of IHCPs. The details on selection of the sampling frame have been published elsewhere [35]. In total, 50 IHCPs who practices within the sampling frame were enlisted (Appendix A). Twenty IHCPs agreed to participate in the study, eight of which dropped out within 2–3 days of data collection. Therefore, 12 IHCPs continued for 18 months, giving us the response rate of 60%. Before starting the study, a questionnaire was used to collect the following basic demographic information from the participating IHCPs: age, gender, duration of practice, and number of outpatients per day for each participating IHCP (Appendix A). The mean (SD) of age and years of work experience of the participating (*n* = 12) IHCPs was 44 (6.9) years and 19.8 (8.2) years, respectively (Appendix A). No significant variation was reported in the mean age and mean year of work experience between the participating and non-participating IHCPs (*n* = 38) (mean (SD) of age and years of work experience of 41.38 (6.6) and 18.8 (10), respectively) (Appendix A).

### 4.3. Data Collection

The IHCPs usually directly dispense the medicine available to them to the patients and do not write prescriptions for the patients. Therefore, in order to have documentation on the medicines prescribed by the participating IHCPs, they were provided with customized prescription pads containing 200 sheets to record prescriptions provided to outpatients in duplicate. The original copy of the prescription was given to the patient, and the duplicate copy was used in this study. Atleast 100 prescriptions per IHCP per month were collected [35]. The prescription included the date of prescription, patient data (age, sex, contact number, and village name), information on the presenting illness or diagnosis, and details of treatment prescribed (brand/generic name, strength, dose, and duration; Appendix A). All prescriptions were recorded, irrespective of whether they contained antibiotics. The IHCPs were asked to provide the details of the presenting complaints or diagnosis, which could be either single or multiple. The data were collected over 18 months from April 2015 to September 2016. Once a month, a research assistant visited the IHCPs to collect the completed prescription pads and once weekly to check and collect any missing data. The diagnoses made by the IHCPs were not externally validated, and the presenting complaints reported by the IHCPs were considered final. The IHCPs were informed that the primary objective of the study was to assess prescribing in general.

### 4.4. Data Management

Data from customized prescription pads were entered using the Microsoft.NET Framework 4.0 software (Microsoft Corp. Redmond, WA, USA) and analyzed using Stata14 (Stata Corp. College Station, TX, USA). Before analysis, the data were checked for missing information. Incomplete prescriptions with missing information on age, sex, and/or presenting complaints/diagnoses were not included in the final analysis.

### 4.5. Data Analysis

Descriptive statistics were calculated for the total number of prescriptions collected and their distribution according to the age and sex of the patient and the season. The IHCPs were asked to write down all presenting complaints for a given patient. The presence of single or multiple presenting complaints/diagnoses provided a large matrix of prescription patterns. We got 323 different combinations of complaints. To simplify the analysis of antibiotic prescription patterns, presenting complaints were arranged in a descending order of frequency for one, two, and more than two presenting complaints. Antibiotic prescription patterns for different combinations of presenting complaints were explored and analyzed. Prescription patterns were presented in the results, if the frequency of the presenting complaint was higher than 100. Because the IHCPs prescribed antibiotics under both generic and trade names, all the trade names of antibiotics were manually converted to their generic names. Each prescribed antibiotic was coded according to the World Health Organization (WHO) Collaborating Centre for Drug Statistics Methodology, Anatomical Therapeutic Chemical (ATC) classification with the DDD, according to the fifth level of the ATC classification, J01 (antibacterial for systemic use) [36]. All DDDs were calculated per thousand prescriptions per presenting complaint/diagnosis (TPP) or a combination of presenting complaints/diagnoses (DDD/TPP) [23,37,38,39,40,41]. FDC were assigned DDDs based on the average use of combinations, without considering and comparing the strengths of various components [36]. Prescription patterns were examined in three age groups (≤5, 6–17, and ≥18 years) and over four seasons. Seasons were divided as follows: pre-monsoon season (March to May), monsoon season (June to September), post-monsoon season (October to December), and winter season (January and February), with average maximum temperatures of 35 °C–40 °C, 29 °C–31 °C, 27 °C–31 °C, and 20 °C–24 °C, respectively [42]. Because data were collected over 18 months (April 2015 to September 2016), the study covered six seasons (two pre-monsoons, two monsoons, one post-monsoon, and one winter). Proportions with their 95% CI were calculated. The frequency and percentage were calculated for categorical variables and the median and inter quartile range were calculated for continuous (numerical) variables.

A heat map was generated to study the clustering of antibiotic prescribing according to diagnosis (column) and for each prescriber (rows 1 to 12) (Appendix A). Antibiotic prescribing was pseudo-colored using the heat map feature in GraphPad Prism 7.03 (GraphPad Software, San Diego, CA). Statistical analysis was performed using GraphPad Prism 7.03 software (GraphPad Software, San Diego, CA). For a comparison of variance in antibiotic prescribing rate by 12 IHCPs based on different diagnosis, two-way ANOVA with multiple comparisons tests were performed and F-ratios were calculated for rows (prescriber) and columns (diagnosis) [43].

### 4.6. Ethical Approval

The Institutional Ethics Committee (IEC) of RDGMC, Ujjain, approved the study (approval number DNR-311/2013). Before the start of the study, the IHCPs were explained the aim of the study and were assured of confidentiality. The institutional ethics committee approved the use of verbal informed consent as the participants were not ready to give the consent in written form.

## 5. Conclusions

The results of this study reveal that the IHCPs prescribe antibiotics more commonly than other medications for common illnesses in both children and adults. Fluoroquinolones and third-generation cephalosporins were the antibiotics most commonly prescribed by IHCPs for the common illnesses analyzed in the study.

## Figures and Tables

**Figure 1 antibiotics-08-00139-f001:**
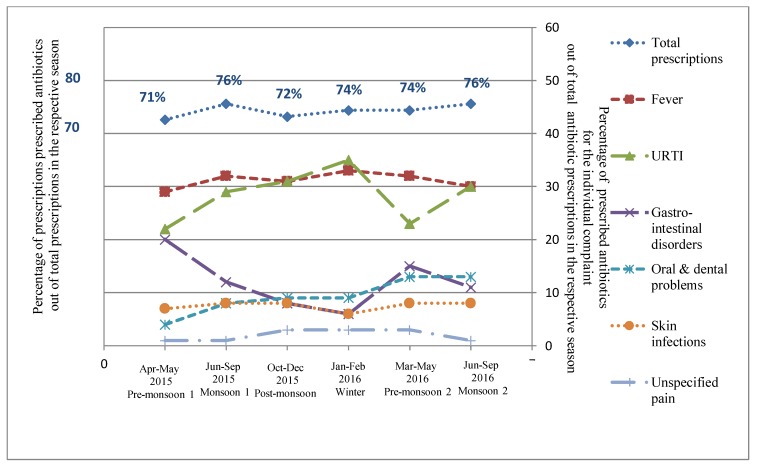
Relative distribution of proportion of prescribed antibiotics out of the total prescriptions in the respective seasons and percentage ofprescribed antibiotics for the individual complaint out of the total antibiotic prescriptions in the respective seasons in rural Ujjain, India. X axis represents the seasons (April–May 2015 Pre-monsoon 1, June–September 2015 Monsoon 1, October–December 2015 Post-monsoon, January–February 2016 Winter, March–May 2016 Pre-monsoon 2, June–September 2016 Monsoon 2); Primary Y axis on left represents the percentage of prescribed antibiotics out of total prescriptions in the respective season and secondary Y axis on right represents the percentage of prescribed antibiotics for the individual complaints (Fever, URTI—Upper respiratory tract infection, Gastro-intestinal disorders, Oral and dental problems, Skin infections, and Unspecified pain) out of the total antibiotic prescriptions in the respective seasons.

**Figure 2 antibiotics-08-00139-f002:**
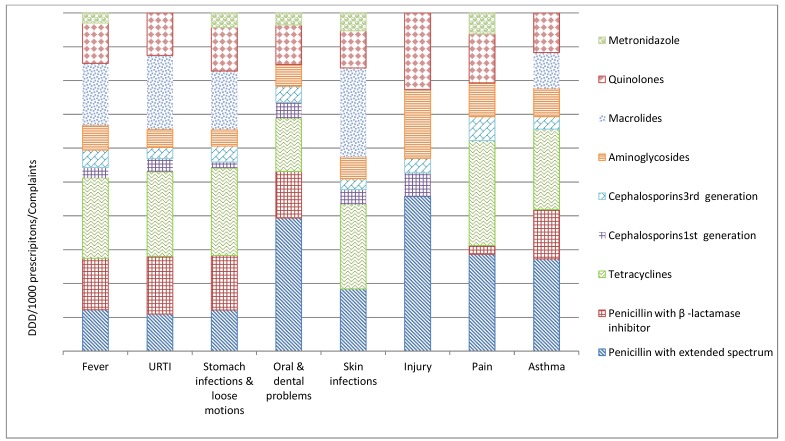
Relative distribution of defined daily doses/1000 prescriptions/presenting complaints (DDD/TPP) in rural Ujjain, India. X axis represents the defined daily doses/1000 prescriptions/presenting complaints (DDD/TPP); Y axis represents the complaints (URTI—Upper respiratory tract infection, Pain—Unspecified pain).

**Figure 3 antibiotics-08-00139-f003:**
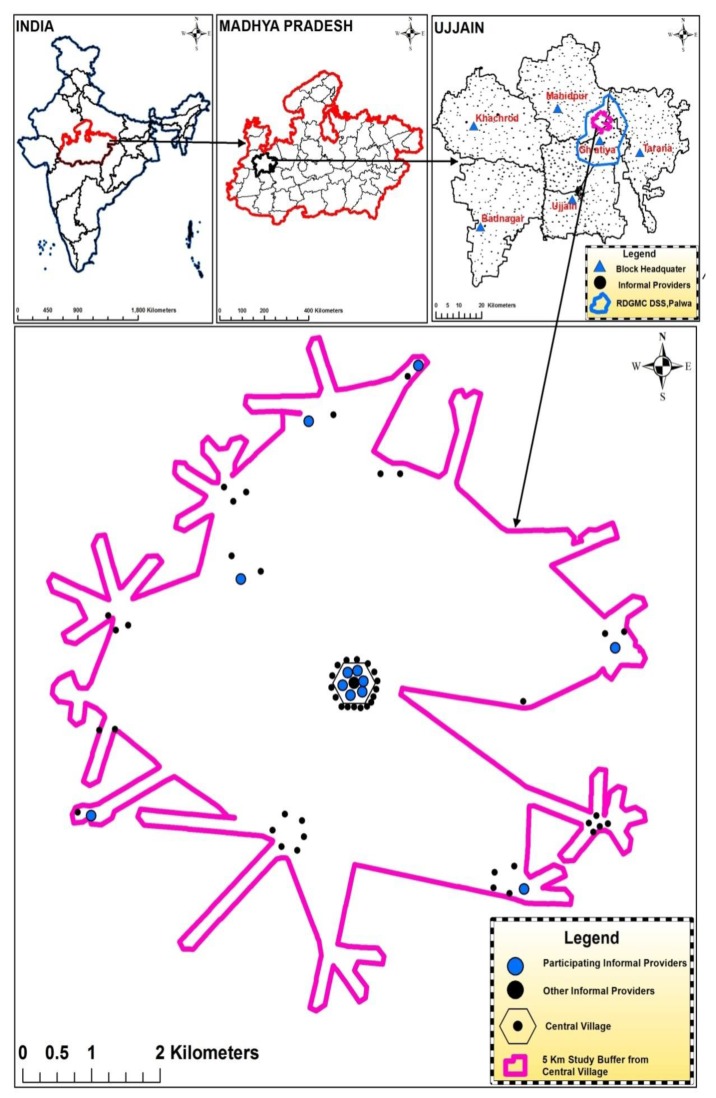
Geographical location of the selected informal healthcare providers. The maps show India, Madhya Pradesh, Ujjain district, and the location of participating informal healthcare providers within the sampling frame.

**Table 1 antibiotics-08-00139-t001:** Incidence of antibiotic prescribing amongst the prescriptions collected from informal healthcare providers in rural Ujjain.

Co-variates	Total Prescriptions *N*	Antibiotic Prescriptions *n*	Proportions of Antibiotic Prescriptions (*n*/*N*)	95% Confidence Interval
	15,322	11,336	0.74	0.73–0.75
**Age groups**				
18 years and above	11,223	7996	0.71	0.70–0.72
6–17 years	1852	1441	0.78	0.76–0.80
0–5 years	2247	1899	0.85	0.83–0.86
**Gender**				
Female	6643	4829	0.73	0.72–0.74
Male	8679	6507	0.75	0.74–0.76
**Presenting complaints**				
Fever	4118	3569	0.87	0.86–0.88
URTI **	4047	3273	0.81	0.79–0.82
Gastro-intestinal disorders	2107	1271	0.60	0.58–0.62
Oral & dental problems	1273	1126	0.88	0.87–0.90
Skin infections	1068	844	0.79	0.76–0.81
**Seasons**				
Pre-monsoon 1	1329	938	0.71	0.68–0.73
Pre-monsoon 2	2432	1811	0.74	0.73–0.76
Monsoon 1	2869	2166	0.75	0.74–0.77
Monsoon 2	3100	2350	0.76	0.74–0.77
Post-monsoon	3423	2476	0.72	0.71–0.74
Winter	2169	1595	0.74	0.72–0.75

** URTI—Upper respiratory tract infections.

**Table 2 antibiotics-08-00139-t002:** Number of antibiotics per prescription, fixed dose combinations, and route of administration of antibiotics for the prescriptions collected from informal healthcare providers in rural Ujjain.

Variable	Number of Prescriptions	% of Total
**Number of antibiotics per prescription**	***N* = 11336**	
1	7072	62
2	4046	36
3 and more	218	2
**Number of fixed dose combinations present**	***N* = 2522**	
Ampicillin, Cloxacillin	1552	62
Sulfamethoxazole, Trimethoprim (Co-trimoxazole)	362	14
Norfloxacin, Tinidazole	320	13
Others	288	11
**Route of drug administration**	***N* = 15472**	
Oral (tablets/capsules/syrup)	10128	65
Parenteral	5007	32
Others	337	3

**Table 3 antibiotics-08-00139-t003:** Number of prescriptions by Anatomical Therapeutic Chemical classification antibiotic category/route of administration (O-Oral/P-Parenteral) and complaints/age group in rural Ujjain, India.

Age Groups & Complaints		Antibiotics																	
		Penicillin with Extended Spectrum; Amoxicillin and Ampicillin with Cloxacillin; J01CA *n* = 4120		Penicillin with β -Lactamase Inhibitor, Amoxicillin with Clavulanic acid; J01CR *n* = 20		Tetracyclines; J01A *n* = 323		Cephalosporins; J01D				Aminoglycosides; J01GB *n* = 1932		Macrolides; J01FA *n* = 157		Fluoroquinolones; J01MA *n* = 4307		Metronidazole; P01AB *n* = 147	
								1st Generation; J01DB *n* = 90		3rd Generation; J01DD *n* = 3062									
		O	P	O	P	O	P	O	P	O	P	O	P	O	P	O	P	O	P
**Fever**	0–5yr	315	9	-	-	-	-	10	-	25	148	-	39	5	-	91	-	-	-
	6–17 yr	191	20	-	-	2	-	3	-	30	143	-	141	2	-	140	2	-	-
	18 yr and above	357	59	5	-	95	-	5	1	226	623	-	644	13	-	1657	53	-	1
**URTI***	0–5yr	747	39	2	1	1	-	10	-	118	315	-	48	10	-	86	-	-	-
	6–17 yr	326	39	-	-	10	-	12	-	42	222	-	92	6	-	114	1	-	-
	18 yr and above	463	121	3	-	76	-	9	-	96	577	1	222	42	-	843	11	-	-
**Gastro-** **intestinal disorders**	0–5yr	52	3	-	-	-	-	2	-	1	21	-	79	-	-	20	-	3	-
	6–17 yr	16	4	-	-	-	-	-	-	2	18	-	54	1	-	28	1	9	2
	18 yr and above	47	9	3	-	38	-	-	1	12	169	-	381	1	-	311	13	80	48
**Oral &** **dental problems**	0–5yr	3	-	-		-	-	-	-	-	-	-	-	-	-	-	-	-	-
	6–17 yr	5	1	-	-	-	-	-	-	-	-	-	1	-	-	8	-	-	-
	18 yr and above	872	55	4	-	7	-	4	-	3	19	-	39	-	-	185	1	1	1
**Skin infections**	0–5yr	33	12	-	-	1	-	6	-	5	10	-	7	-	-	10	-	-	-
	6–17 yr	42	18	-	-	5	-	7	-	4	26	-	35	12	-	42	-	-	-
	18 yr and above	62	31	-	-	54	-	10	-	12	48	-	72	64	1	289	-	1	-
**Injury**	0–5yr	9	1	-	-	-	-	5	-	1	2	-	3	-	-	3	-	-	-
	6–17 yr	10	9	-	-	-	-	2	-	1	15	-	5	-	-	23	-	-	-
	18 yr and above	33	18	-	-	-	-	4		1	43		37	-	-	163	1	-	-
**Pain**	0–5yr	-	1	-	-	-	-	-	-	-	-	-	2	-	-	1	-	-	-
	6–17 yr	-	-	-	-	-	-	-	-	-	-	-	3	-	-	1	-	-	-
	18 yr and above	26	2	-	1	6	-	-	-	7	13	-	12	-	-	172	-	1	-
**Asthma**	0–5yr	3	1	-	-	-	-	-	-	1	2	-	-	-	-	-	-	-	-
	6–17 yr	1	1	-	-	-	-	-	-	-	2	-	-	-	-	1	-	-	-
	18 yr and above	41	13	1	-	28	-	-	-	10	49	-	15	4	-	30	6	-	-

*URTI—Upper respiratory tract infections; O—Oral; P—Parenteral.

## Data Availability

Due to ethical and legal restrictions, all inquiries should be directed to The Chairman, Ethics Committee, R.D. Gardi Medical College, Agar Road, Ujjain, India 456006 (Emails: iecrdgmc@yahoo.in, uctharc@bsnl.in), giving all details of the publication. Upon verification of the genuineness of the inquiry, the data will be made available. For reference, please quote ethical permission no. 311, dated July 17, 2013.

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
