# Peer review of "Antibiotic Prescribing by Informal Healthcare Providers for Common Illnesses: A Repeated Cross-Sectional Study in Rural India"

_antibiotics, 2019, doi:10.3390/antibiotics8030139_

Round 1

Reviewer 1 Report

In their cross-sectional study, Shweta Khare and colleagues described the characteristics of antibiotic prescribing by informal healthcare providers (IHCP) for common illnesses in a region of rural India. In my opinion, the topic is of interest, both in general and for readers of Antibiotics. Below are some major and minor comments on the technical merits of the paper.

Major comments

1) The authors concluded that they observed excessive antibiotic prescribing for common illnesses by IHCP. While this is likely true, particular attention should be devoted to denominators (no only for this conclusion but for all the study results and related interpretations). Indeed, according to the methods, the study population is not made of patients with common illnesses, but by patients with common ilnesses receiving prescriptions. This difference is important, since it touches on core aspects of the interpretation of results. For example, a more correct conclusion would likely be “for common illnesses, patients were prescribed more commonly antibiotics than other medications”. In other words, how large was the proportion of patients with common illnesses who did not receive prescriptions (who is not included in the denominator)? If different from zero, this would change the interpretaion of results

2) The previous comment also applies to logistic regression models. For example, males received more antibiotics than other drugs compared to females. However, if the proportion of males not receiving treatments was larger than the proportion of females not receiving treatment, this could mean that, overall, males received less antibiotic than females for common illnesses.

3) The use of logistic regression with dummy variables may be not very useful here. For example, why using skin infections as a reference? I would find more interesting just to report the proportion of treated patients for each condition with its CI. In addition, how were combined complaints categorized?

4) Independent of the model used for exploring associations, there are repeated measures which are inherently non-independent. Considering the levels of the variable, the authors may consider random effects for accounting for inter-IHCP variability

Minor comments

1) Table 3 is not easy to read, preventing the readers for rapidly comparing possible differences.

2) In my opinion, this information may be better suited for graphs. It would be interesting to adjust cells of the heatmap for numbers of total prescriptions as denominators

Reviewer 2 Report

This manuscript discusses an important global health issue, namely the  practice of prescribing antibiotics indiscriminately, even in the absence of bacterial infection. The data support the authors' conclusions and are quite extensive. I only have a few trivial corrections or comments.

The words "healthcare" and "health care" are both used in the manuscript. Please pick one or the other and be consistent. "Healthcare" is probably preferable. Is the key in Figure 1 incomplete? I don't see definitions for all of the curves. In the Figure 2 legend, are the definitions for the X and Y axes switched?

Round 2

Reviewer 1 Report

Thank you for your kind and scientifically sounded responses to comments.